# Feeding Essential Oils to Neonatal Holstein Dairy Calves Results in Increased Ruminal Prevotellaceae Abundance and Propionate Concentrations

**DOI:** 10.3390/microorganisms7050120

**Published:** 2019-05-01

**Authors:** Prakash Poudel, Kelly Froehlich, David Paul Casper, Benoit St-Pierre

**Affiliations:** 1Department of Animal Science, South Dakota State University, Animal Science Complex, Box 2170, Brookings, South Dakota, SD 57007, USA; Prakash.Poudel@sdstate.edu; 2Department of Agricultural Sciences, Lincoln University, Lincoln 7647, Canterbury, New Zealand; Kelly.Froehlich@lincoln.ac.nz; 3Furst-McNess Company, 120 East Clark Street, Freeport, IL 61032, USA; david.casper@mcness.com

**Keywords:** rumen, microbiome, bacteria, essential oils, propionate, *Prevotella*

## Abstract

Since antibiotic use in animal production has become a public health concern, great efforts are being dedicated to find effective and viable alternatives. While essential oils (EO) have become attractive candidates for use in the livestock industry, their mode of action and microbial targets in food animals remain largely uncharacterized. To gain further insight, we investigated the rumen environment of neonatal calves fed calf starter pellets and milk replacer supplemented with a commercial blend of EO. Propionate concentrations were not only found to be higher in EO-fed calves compared to controls (*P* < 0.05), but ruminal bacterial communities also differed greatly. For instance, the abundance of Firmicutes was significantly lower in samples from EO-fed calves than in controls, which appeared to be mostly due to lower Lachnospiraceae levels (*P* < 0.05). In contrast, Bacteriodetes were more abundant in EO-fed calves compared to controls, which was primarily the result of higher Prevotellaceae (*P* < 0.05). Notably, two bacterial operational taxonomic units (OTUs) were significantly more abundant in EO-fed calves; SD_Bt-00966 was found to be a close relative of *Prevotella ruminicola* (97%), while SD_Bt-00978 likely corresponded to an uncharacterized species of Gammaproteobacteria. In addition, Pearson correlation and canonical correspondence analyses revealed potential associations between other ruminal bacterial OTUs and either short chain fatty acids (SCFA) parameters or metrics for calf growth. Together, these results support that EO supplementation in growing dairy calves can modulate rumen function through SCFA production and growth of specific rumen bacterial groups.

## 1. Introduction

Antibiotics have traditionally been used in dairy calf production to increase immunity as well as reduce stress and susceptibility to pathogens. They have been shown to improve rumen development, increase growth performance, reduce neonatal diarrhea, and decrease the risk of calf mortality [1]. However, due to the increased incidence of bacterial resistance and potential risks for food security, antibiotic use in animal production has become a concern, leading to stricter regulation for this practice in the livestock sector. Indeed, policies such as the European Union ban on the use of antibiotics and ionophores in animal production, as well as the phasing out of prophylactic treatments for food animals produced in the USA by the Food and Drug Administration (FDA), have created an urgent need for alternatives. To be viable, these not only have to promote animal welfare, but also optimize animal production, while posing only minimal risks to human health and the environment [2,3]. 

Since antibiotics act as selection agents that ultimately affect the composition of host microbiomes, a common strategy to identify effective alternatives has been to explore the potency of other types of antimicrobials, which have included essential oils (EO), a group of plant secondary metabolites that can be extracted by distillation. As a group, they are very diverse in chemical structure and biological effects, with terpenoids and phenylpropanoids representing the most commonly found types of EO [4]. Studies carried out in ruminants have provided evidence that EO could be used instead of antibiotics for improving animal productivity [5]. While certain reports found no discernable effects of EO supplementation on production or ruminal parameters [6,7], perhaps as a result of dosage or nature of the active compounds, other studies were successful in uncovering positive responses. Their reported effects on rumen function include inhibition of deamination and methanogenesis, resulting in lower ammonia nitrogen and methane, respectively [8]. EO can also reduce ruminal acetate levels, while maintaining total short chain fatty acid (SCFA) production through increased propionate and butyrate production in the rumen [9]. Positive effects of EO supplementation for dairy calf performance have also been reported, such as increased starter feed intake and improved feed efficiency [10]. The benefits of EO on the performance of young ruminants are of particular interest, as they may be the result of changes in the gut microbiome caused by their antimicrobial activities. Furthermore, since the composition of gut microbial communities in neonatal and young animals tends to fluctuate until it becomes stably established later on in life [11], it is more likely to be responsive to manipulation during these early growth stages. 

Ultimately, the purpose of modulating early gut microbiome composition would be to provide long term benefits to the performance and health of adult animals [12]. However, the impact of EO supplementation on the microbiome of young ruminants remains largely unexplored. To gain further insight, we took advantage of a companion study to investigate the rumen environment of dairy calves fed a commercial blend of EO [13]. In this context, the main objective of the investigation presented in this report was to determine the effects of EO supplementation on ruminal bacterial communities. Comparative analyses of ruminal SCFA profiles and bacterial community composition performed between the two dietary groups indicated that supplementation of a standard dairy calf diet (i.e., milk replacer and pelleted calf starter) with the EO product resulted in an increase in rumen propionate concentration that was associated with profound differences in bacterial composition, which included the enrichment of specific uncharacterized bacteria. 

## 2. Materials and Methods 

### 2.1. Sample Collection

The analyses described in this report were performed on samples collected during a previously reported companion study [13], which was conducted at the South Dakota State University (SDSU) Animal Research Wing (ARW; Brookings, SD), with all procedures approved by the SDSU Institutional Animal Care and Use Committee before the start of the trial. For the purpose of the microbiome study presented in this report, rumen fluid was sampled from a subset of 10 of the animals fed milk replacer supplemented with EO (3.75g/feeding, mixed with the milk replacer), and from 10 of the animals fed milk replacer without supplementation. The commercial supplement, manufactured by Ralco, Inc. (Stay Strong for Dairy Calves; Marshall, MN), was a blend of EO (carvacrol, caryophyllene, p-cymene, cineole, terpinene, and thymol) that also included arabinogalactans, a type of hemicellulose known to enhance immune function [14]. All calves were housed individually, fed milk replacer (24%:20% crude protein:fat; as-fed basis), and had ad libitum access to water and pelleted calf starter during the trial (see Appendix A for calf starter ingredient composition). Calves were fed colostrum for the first two days after birth. Milk replacer (with or without EO supplementation, according to treatment group) was then offered by bucket feeding twice every day from d3 until d35, then reduced to once every day starting at d36 to facilitate weaning at d42. From week 1 until week 6, calf starter intake increased from approximately 2% of total dry matter intake (DMI) to approximately 70%; the average daily DMI in the week prior to sampling was 1.38 kg/d for control calves and 1.45 kg/d for EO-fed calves (*P* > 0.05) [13]. The amount of EO supplemented per feeding remained constant during the trial, and it was not adjusted as a function of calf growth or increased DMI. Rumen samples were collected one day after weaning (day 43) from each animal by stomach tubing, with rinsing of the sampling equipment with warm water between each collection. Separate samples (50 mL/sample) were collected for microbiome and SCFA analysis, with the latter supplemented with 25% metaphosphoric acid (W/V) at a ratio of 4:1 before freezing. All samples were stored at –20 °C until analyzed.

### 2.2. SCFA Analysis

Rumen samples mixed with metaphosphoric acid were thawed, then centrifuged to remove particulate (16,000× *g*, 1 min). For each sample, 800 μL of supernatant was mixed with 200 μL of an internal standard (2-ethyl butyric acid, 20 mM). Following injection, SCFAs were separated by gas liquid chromatography (Trace 1310, Thermo Scientific, Bellefonte, PA, USA) on a 0.25 mm i.d. × 30 m capillary column with 0.25 μm film thickness (NukolTM, Supelco Inc., Bellefonte, PA, USA). The injector port temperature was 200 °C, with a split ratio of 100:1, and a column flow of He at a rate of 0.8 mL/min. After starting at 140 °C for a duration of 9.5 min, the oven temperature was increased at a rate of 20 °C/min until it reached 200 °C, at which point it was maintained for 1 min. Detection was completed using a flame-ionization detector with a temperature of 250 °C. Data was analyzed by the software Chromeleon 7.2 CDS, with SCFA concentrations measured based on peak height. For calibration, a mixture of standards (Supelco Volatile Free Fatty Acid Mix 46975, Supelco Inc., Bellefonte, PA) was first analyzed for identification of SCFAs peaks, with 2-ethylbutyric acid serving as an internal standard. External calibration was performed using three different SCFA concentration levels, each measured twice. 

### 2.3. Microbial DNA Isolation and PCR Amplification

Microbial DNA was isolated from rumen samples using a repeated bead beating plus column method [15]. The V1–V3 region of the bacterial 16S rRNA gene was PCR-amplified using the 27F forward [16] and 519R reverse [17] primer pair. PCR reactions were performed with the Phusion Taq DNA polymerase (Thermo Scientific, Waltham, MA, USA) under the following conditions: Hot start (4 min, 98 °C), followed by 35 cycles of denaturation (10 s, 98 °C), annealing (30 s, 50 °C) and extension (30 s, 72 °C), then ending with a final extension period (10 min, 72 °C). PCR products were separated by agarose gel electrophoresis, and amplicons of the expected size (~500 bp) were excised for gel purification using the QiaexII Gel extraction kit (Qiagen, Hilden, Germany). For each sample, approximately 400 ng of amplified DNA were submitted to Molecular Research DNA (MRDNA, Shallowater, TX, USA) for sequencing with the Illumina MiSeq 2 × 300 platform to generate overlapping paired end reads. Briefly, libraries for 16S rRNA amplicons were prepared using overhanging primers targeting the 27F and 519R recognition sequences; the primer overhangs encoded barcodes for sample indexing as well as adapter sequences. MiSeq 2 × 300 sequencing was performed using the MiSeq Reagent Kit v3 following the manufacturers specifications.

### 2.4. Computational Analysis of PCR Generated 16S rRNA Amplicon Sequences

Unless specified, sequence data analysis was performed using custom written Perl scripts (available upon request). Raw bacterial 16S rRNA gene V1–V3 amplicon sequences were provided by Molecular Research DNA as assembled contigs from overlapping MiSeq 2 × 300 paired-end reads from the same flow cell clusters. Reads were then selected to meet the following criteria: Presence of both intact 27F (forward) and 519R (reverse) primer nucleotide sequences, length between 400 and 580 nt, and a minimal quality threshold of no more than 1% of nucleotides with a Phred quality score lower than 15. 

Following quality screens, sequence reads were aligned, then clustered into operational taxonomic units (OTUs) at a genetic distance cutoff of 5% sequence dissimilarity [18]. While 3% is the most commonly used clustering cutoff for 16S rRNA, it was originally recommended for full length sequences, and may not be suitable for the analysis of specific sub-regions, since nucleotide sequence variability is not constant across the entire length of the 16S rRNA gene. In this context, if 3% is a commonly accepted clustering cutoff for V4 or V4–V5 regions, which are the least variable of the hypervariable regions, then a higher cutoff should be used for the V1–V3 region, since V1 is the most variable region of the 16S rRNA gene. 

OTUs were screened for DNA sequence artifacts using the following methods. Chimeric sequences were first identified with the chimera.uchime and chimera.slayer commands from the MOTHUR open source software package [19]. Secondly, the integrity of the 5′ and 3′ ends of OTUs was evaluated using a database alignment search-based approach; when compared to their closest match of equal or longer sequence length from the NCBI nt database, as determined by BLAST [20], OTUs with more than five nucleotides missing from the 5′ or 3′ end of their respective alignments were discarded as artifacts. Single read OTUs were subjected to an additional screen, where only sequences that had a perfect or near perfect match to a sequence in the NCBI nt database were kept for analysis (i.e., that the alignment had to span the entire sequence of the OTU, and a maximum of 1% of dissimilar nucleotides was tolerated). 

After removal of sequence chimeras and artifacts, taxonomic assignment of valid OTUs was determined using a combination of RDP Classifier [21] and BLAST [20]. The List of Prokaryotic Names with Standing in Nomenclature (LPSN—http://www.bacterio.net) was also consulted for information on valid species belonging to taxa of interest [22].

### 2.5. Computational Analysis for Alpha and Beta Diversity

Using custom Perl scripts, all datasets were randomly rarefied to 1800 reads, which were then used to create “shared”-type formatted files. All subsequent steps were performed using commands in MOTHUR [19]. Chao1, Shannon, and Simpson indices, as well as observed OTUs and coverage, were determined from the shared files using summary.single. For principal coordinate analysis (PCoA), Bray–Curtis distances were first determined using summary.shared, which were then used as input for the command pcoa. Principal Components 1 (PC1) and 2 (PC2), representing the highest levels of variation, were plotted using Microsoft Excel. Canonical correspondence analysis (CCA) was conducted in R (version 3.2.3) using the command cca from the vegan package (version 3.2.5), with outputs plotted using the command plot. 

### 2.6. Statistical Analyses

An unpaired *t*-test was used to compare rumen SCFAs levels as well as the abundance of bacterial taxonomic groups, respectively, between samples from calves fed the EO supplemented diet and calves fed the control diet. The *t*-test was conducted using the online GraphPad Software (https://www.graphpad.com/quickcalcs/ttest1.cfm). Pearson correlation coefficients and associated *P* values were calculated using Microsoft Excel. Means of two groups were considered to be significantly different when *P* ≤ 0.05.

### 2.7. Accession Numbers for Next Generation Sequencing Data

Raw sequence data are available from the NCBI Sequence Read Archive under Bioproject PRJNA475807 and SRA accession SRP150434. 

## 3. Results

### 3.1. Comparative Analysis of Ruminal SCFA between EO Supplemented and Non-Supplemented Diets 

Ruminant animal performance is dependent on ruminal SCFA production. Amongst the SCFAs analyzed (Figure 1), propionate was found in higher concentration (*P* < 0.05) in the rumen of EO-fed calves (40.25 ± 3.03 mM) compared to calves fed the control diet (31.06 ± 3.14 mM). While numerically greater in animals on the EO-supplemented diet, acetate, valerate, and total SCFA concentrations were not found to be statistically different between the two diets. A numerical difference in the acetate: propionate ratio was observed between the two treatments (EO: 1.24 ± 0.04; control: 1.38 ± 0.06; *P* = 0.072).

### 3.2. Effects of EO on the Taxonomic Composition of Ruminal Bacteria in Growing Calves

From the 20 samples analyzed, a total of 347,254 high quality sequence reads were generated, with an average of 16,376 ± 4472 per sample. Taxonomic analysis identified six phyla across all samples, with Firmicutes, Bacteriodetes, and Proteobacteria being the most highly represented (Table 1, Figure 2). The relative abundance of Firmicutes was significantly lower (*P* < 0.05) in EO-fed calves (43.68% ± 6.92%) compared to controls (73.22% ± 6.79%), which appeared to be mostly due to lower Lachnospiraceae levels in EO-fed samples (*P* < 0.05). In contrast, Bacteriodetes were more abundant in EO-fed calves (44.63% ± 6.28%) compared to controls (13.45% ± 6.02%), which was primarily the result of higher Prevotellaceae (44.20% ± 6.27% vs 9.70% ± 5.94%) (*P* < 0.05). Proteobacteria, mostly represented by unclassified Gammaproteobacteria, were also found to be significantly higher (*P* < 0.05) in samples from EO-fed calves compared to control calves (3.49% ± 1.32% vs 0.17% ± 0.13%, respectively). 

### 3.3. Effects of EO on the Ruminal Bacterial Community Structure in Growing Calves 

To gain further insights into the community level compositional differences between EO and control ruminal environments, alpha and beta diversity analyses were conducted. Diversity of ruminal bacteria was not affected by treatment with EO under these conditions, since no statistical differences were observed for Chao1, Simpson, and Shannon indices (Table 2). However, a beta diversity analysis using principle coordinate analysis (PCoA) based on operational taxonomic unit (OTU)-level Bray–Curtis dissimilarity (Figure 3) supported that the composition of EO and control samples were different from each other, as their respective data points were not evenly distributed between clusters.

From a total of 4154 OTUs that were identified in this study, 31 OTUs were designated as main OTUs, which were defined as having a mean relative abundance of at least 1% for at least one treatment (Table 3 and Table 4 and Appendix A). As a group, main OTUs represented 68.9% and 67.0% of sequence reads in EO and control fed samples, respectively. Only four main OTUs (SD_Bt-00966, SD_Bt-00967, SD_Bt-00986, and SD_Bt-36860) were found to have a sequence identity of 95% or greater to their closest valid relative, indicating that at least 27 main OTUs likely corresponded to uncharacterized ruminal bacterial species.

Overall, main OTUs showed a phylogenetic distribution reflecting their respective treatment, with higher representation of Bacteriodetes-affiliated OTUs in EO-fed calves and higher abundance of Firmicutes OTUs in control calves. Five of the Bacteriodetes affiliated OTUs were found in higher abundance in EO-fed calves compared to control, and were affiliated to the genus Prevotella (Table 3). Of these, only SD_Bt-00966 showed significantly higher relative abundance in calves fed EO (19.51% ± 5.32%) compared to control (2.69 % ± 1.80%). Its closest known relative was identified as *Prevotella ruminicola* (97% sequence identity). Firmicutes included by far the highest number of main OTUs, but none showed statistical differences based on treatment (Table 4). OTU SD_Bt-00179 was observed in greater abundance in EO samples (13.6×), but these differences were not found to be statistically significant. OTUs SD_Bt-00125, SD_Bt-00732, SD_Bt-00975, SD_Bt-00980, SD_Bt-00983, SD_Bt-00998, and SD_Bt-36860 were found to be between 10× to 75.2× greater in control-fed calves compared to EO-fed calves. While these differences could help explain the higher abundance of Firmicutes in control-fed calves, they were not supported by statistical analysis. Most Proteobacteria were represented by a single OTU (SD_Bt-00978), which was higher (*P* < 0.05) in EO-fed calves compared to controls (3.44% ± 1.30% vs 0.17% ± 0.13%). The most abundant Actinobacteria OTU (SD_Bt-00967) was numerically lower in EO-fed calves, but by only a 2.5× difference with controls. Based on its high sequence identity to its closest valid relative, this OTU may have represented a strain of *Olsenella umbonata* (Table 2).

### 3.4. Identification of Potential Associations between Main OTUs and Ruminant Performance Parameters

To explore potential associations between dairy calf performance parameters and ruminal bacterial OTUs, two approaches were used. First, canonical correspondence analyses were performed using SCFAs levels and growth parameters as explanatory variables, respectively (Figure 4). Based on the length of the arrows for the SCFA biplot, which is indicative of the respective strength of association of the explanatory variables, the acetate propionate ratio, total SCFAs, as well as the respective levels of propionate, acetate, and iso-butyrate, were found to display, overall, the strongest associations with OTUs. SD_Bt-00179 uniquely showed high correspondence to multiple SCFA attributes (total SCFAs, acetate, and propionate), while other OTUs appeared more strongly associated with individual SCFA conditions, such as observed for the acetate–propionate ratio (SD_Bt-00125, SD_Bt-00975, and SD_Bt-00009). While butyrate did not show as strong an influence as other SCFAs by this analysis, CCA indicated a strong association between butyrate and SD_Bt-00732. When calf growth performance parameters were used as explanatory variables, body length and heart girth showed the strongest correspondence with OTUs. SD_Bt-00009, SD_Bt-30048, and SD_Bt-00070 were found to be more strongly associated with body length, while SD_Bt-00977, SD_Bt-00732, and SD_Bt-00967 were more strongly associated with heart girth.

Based on Pearson correlation coefficient analysis (Appendix A), the following positive correlations were identified. Butyrate concentrations, which are critical to the development of ruminal papillae in growing calves, were strongly associated (*P* < 0.05) with OTUs SD_Bt-00995 (*r* = 0.733) and SD_Bt-00732 (*r* = 0.654), and showed a tendency for correlation with SD_Bt-00992 (*r* = 0.622, *P* = 0.055). Valerate levels were also strongly correlated (*P* < 0.05) with OTU SD_Bt-00995 (*r* = 0.635), and showed a tendency with SD_Bt-00732 (*r* = 0.592, *P* = 0.072). Finally, for SCFA parameters, a tendency for correlation was found for three OTUs with the acetate–proprionate ratio: SD_Bt-00009 (*r* = 0.596, *P* = 0.069), SD_Bt-00070 (*r* = 0.575, *P* = 0.082), and SD_Bt-00718 (*r* = 0.558, *P* = 0.094). Amongst the calf growth parameters tested, the only statistically supported association by Pearson correlation was a tendency between SD_Bt-00978 and hip width (*r* = 0.557, *P* = 0.094).

## 4. Discussion

Development of the ruminal microbiome in neonatal ruminants is a complex and dynamic process involving microbial colonization and succession that ultimately culminates in the establishment of a stable microbial community that can support the host animal by producing SCFAs through fermentation of ingested feed [12]. This stage may provide a window of opportunity for manipulation, potentially allowing to increase the productivity and health of mature host animals through modulating the composition of their developing rumen microbiome [23,24]. While solid feed has so far been found to be the main factor affecting rumen microbiome composition and community structure during pre-ruminal microbial colonization [25], there is growing interest in identifying compounds that could be used as feed additives to improve the rumen function of calves as they mature. Since they exhibit antimicrobial properties, and have shown potential as alternatives to antibiotics, EO have become attractive candidates to serve this purpose [9]. 

For instance, thymol and carvacrol have been found to act as potent antimicrobials against pathogens such as *Escherichia coli*, *Salmonella typhimurium*, *Staphylococcus aureus*, S. *epidermidis*, as well as *Listeria monocytogenes* [26,27]. These compounds were also reported to exhibit antimicrobial activity against ruminal bacteria. Indeed, EO had previously been reported to inhibit the growth of most pure cultures of rumen bacteria [8]. *Clostridium sticklandii* and *Peptostreptococcus anaerobius* were found to be the most sensitive species, while *Streptococcous bovis* was the most resistant. Certain species, such as *P. ruminicola* and *P. bryantii*, could also adapt to grow in the presence of higher EO concentrations. Similarly, Patra and Yu (2012) [28] have found significant reductions in growth of *Fibrobacter succinogenes*, *Ruminococcus flavefaciens*, and *R. albus* in the presence of EO from clove, eucalyptus, garlic, oregano, or peppermint.

In light of the limited available information on the effect of EO on ruminal microbiomes, we took advantage of a dairy calf production trial to investigate the response of ruminal bacterial composition and SCFA levels to dietary supplementation with EO. All dairy calves sampled in this study were on a diet regimen to promote early rumen development with the use of calf starter pellets, which provided a mix of the main substrates that rumen microorganisms would metabolize from a typical solid diet. Milk replacer was offered in buckets once to twice every day for calves to drink from, which not only promoted solid feed consumption, but also minimized formation of the esophageal groove which is induced by suckling. Thus, under this regimen, development of the rumen would have started well prior to weaning, so there was no sudden transition of the rumen from non-functional to functional status. As the main precursor for glucose synthesis, higher propionate levels are generally considered beneficial for ruminant production [29]. Since significantly greater concentrations of propionate were observed in the rumen of EO-fed calves, inclusion of this blend in the diet of young ruminants appeared to create a ruminal environment that was favorable for animal performance. For growing calves, butyrate is typically considered the more desirable SCFA, as it is involved in initiating the development of rumen papillae through stimulation of rumen epithelial metabolism [30]. While propionate may also be used to a lesser extent as a source of energy for rumen papillae development [31], its main effect would more likely be to improve animal performance, through its role as a major substrate for gluconeogenesis, rather than promoting rumen development. 

Ruminal SCFA concentrations and profiles are dependent on the respective composition of the diet and of the host’s ruminal microbial communities [29]. Accordingly, differences in rumen microbial community composition were observed between EO-fed calves and controls. While 16S rRNA-based taxonomic analysis revealed a substantial degree of variation in composition amongst calves in this study, differences between treatments were observed. Notably, the respective abundances of two OTUs, SD_Bt-00966 and SD_Bt-00978, were found to be significantly greater in the rumen of EO-fed calves compared to control calves by a factor of 7.2X and 20.2X. While taking into consideration the level of microbial diversity and functional diversity that exists in the rumen, higher propionate levels in response to EO could involve these OTUs at least in part, perhaps through expression of propionate production pathways [32,33]. While SD_Bt-00978 was phylogenetically too distant from its closest relative to reliably infer function based on its 16S rRNA gene sequence (*Haemophilus influenzae*, 84% sequence identity), SD_Bt-00966 presented a close match to *P. ruminicola* (97% sequence identity). This bacterial species has been defined as a carbohydrate utilizer [34,35] with the ability to tolerate low pH [36]. Interestingly, *P. ruminicola* was reported to be able to grow in the presence of elevated EO [8], and many strains possess the ability to decarboxylate succinic acid to propionic acid [37,38]. While it remains to be determined whether SD_Bt-00966 represented a strain of *P. ruminicola* or if it corresponded to an uncharacterized species of *Prevotella*, these properties make it an interesting candidate to pursue towards linking increases in ruminal propionate to the addition of EO in a diet. 

The predominance of Firmicutes in the rumen of calves fed the control diet is consistent with a number of studies conducted with pre-weaned dairy calves [39,40], while other groups have reported combinations of Bacteriodetes, Firmicutes, and Proteobacteria [11,41]. Notably, Patra and Yu (2015) [42] have observed a lower abundance of Firmicutes combined with higher levels of *Prevotella* in response to phenolic EO extracted from oregano, which is consistent with our observations. The same report also indicated that the effects of EO on rumen bacterial communities were dependent on the chemical nature of the EO provided as supplement. Indeed, the type of EO used, the composition of their active components, as well as their dosage, may affect their ability to modulate performance or the rumen environment [9,43,44]. For instance, a phenolic structure, as well as the presence of hydroxyl groups and their respective position in a compound, can affect the antimicrobial potency of certain EO [45,46]. While the specific modes of action of EO still remain to be determined, they are thought to be more effective in combinations, as different types of compounds may be more likely to affect microbial growth or survival through distinct mechanisms. For instance, additive effects of carvacrol and thymol against *S. aureus* and *Pseudomonas aeruginosa* have previously been reported [47]. As Gram-negative bacteria appear to be less susceptible to the antimicrobial properties of EO compared to Gram-positive, perhaps because of their cell wall structure [45,46], compounds such as p-cymene, which can induce swelling of bacterial cell walls, could act in synergy with other EO components by facilitating their uptake into target cells [48]. Conversely, we would anticipate that bacterial species able to thrive in the presence of EO would possess structural and/or enzymatic adaptations to counter EO antibacterial mechanisms. In the context of the current search for effective and sustainable alternatives to antibiotics, further investigations of EO-resistant bacteria could yield valuable insight into cellular mechanisms that could be targeted by future generations of antimicrobials. As the analyses presented in this report were performed using relative abundance, it also remains to be determined whether the observed changes in composition were the result of increased cell numbers for EO resistant bacterial OTUs or a consequence of growth suppression of susceptible bacterial groups. In addition to their antimicrobial effects, EO can also affect host gut cells, notably by modulating the activity of membrane channels and the uptake of ions [49]; thus it also remains to be determined whether EO modulation of ruminal bacterial communities and host cell activity represent independent effects or can act in synergy.

In addition to EO, the commercial additive used in this study also included arabinogalactans, a type of hemicellulose primarily composed of galactose and arabinose that is intended to enhance immune function [14]. Intriguingly, uncharacterized ruminal spirochete strains that preferably metabolize arabinogalactans over cellulose have been identified [50], indicating the existence of a niche for this substrate in the rumen. However, considering that butyrate was more prominent than propionate in human fecal cultures grown with arabinogalactans [51], these polysaccharides may not be responsible for the increase in ruminal propionate observed in this study. As the effects of arabinogalactans on the ruminal environment and its microbiome remain largely unexplored, future investigations will be necessary to determine if they impact development of the rumen.

## 5. Conclusions

In conclusion, the results of this study support a beneficial effect of EO on the SCFA profile of dairy calves, which may potentially promote increased performance later in life. This report also indicates that at least two ruminal bacterial species belonging to distinct phylogenetic lineages may be upregulated by feeding EO to young calves. Together, these results thus support that EO can effectively be used to modulate the ruminal environment and microbiome of young bovine animals towards potentially improving their nutrition, performance, and health during the productive stages of their life.

## Figures and Tables

**Figure 1 microorganisms-07-00120-f001:**
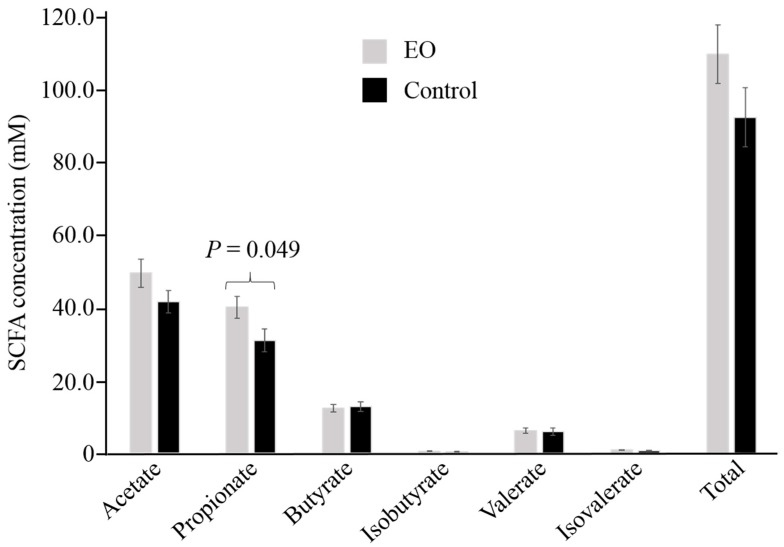
Short chain fatty acid (SCFA) profiles of rumen samples from essential oil (EO)-supplemented and control diet fed calves. Values shown represent the mean and standard error of the means for 10 samples per treatment.

**Figure 2 microorganisms-07-00120-f002:**
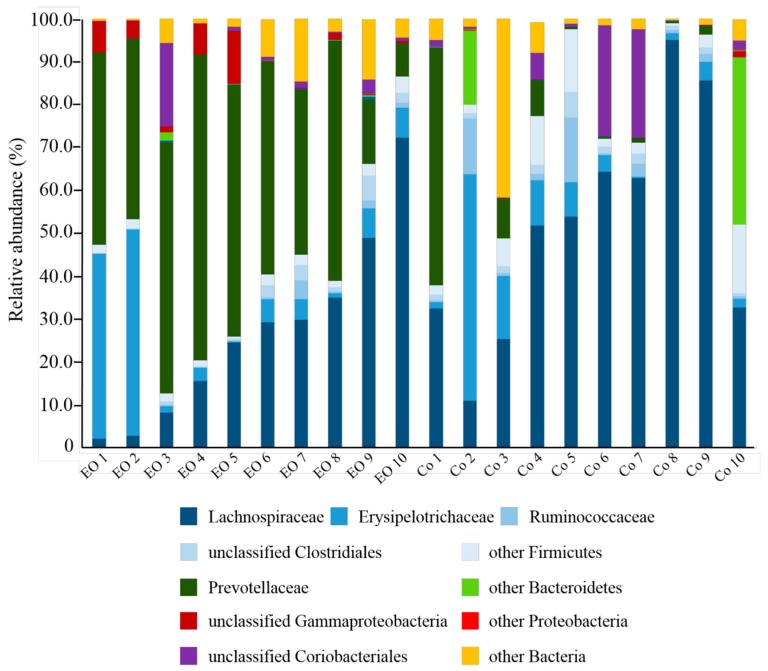
Family level taxonomic composition of rumen bacterial populations in EO-fed calves and controls (Co). Families belonging to the same phylum are represented by different shades of the same color: Firmicutes (blue), Bacteriodetes (green), Proteobacteria (red), and Actinobacteria (purple).

**Figure 3 microorganisms-07-00120-f003:**
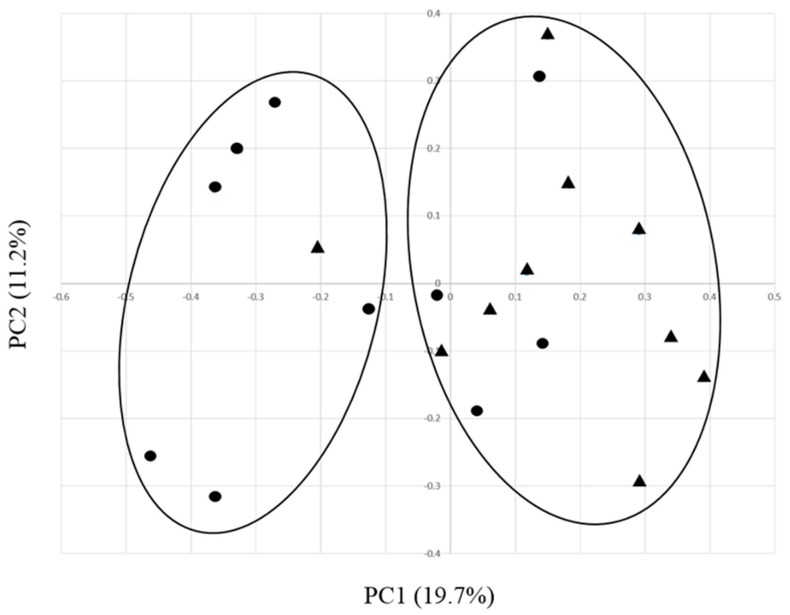
Comparison of rumen bacterial communities from EO-supplemented and control diet fed dairy calves using principle coordinate analysis (PCoA). The x and y axes correspond to Principal Components 1 (PC1) and 2 (PC2), which explained the highest level of variation. EO and control samples are represented by circles and triangles, respectively.

**Figure 4 microorganisms-07-00120-f004:**
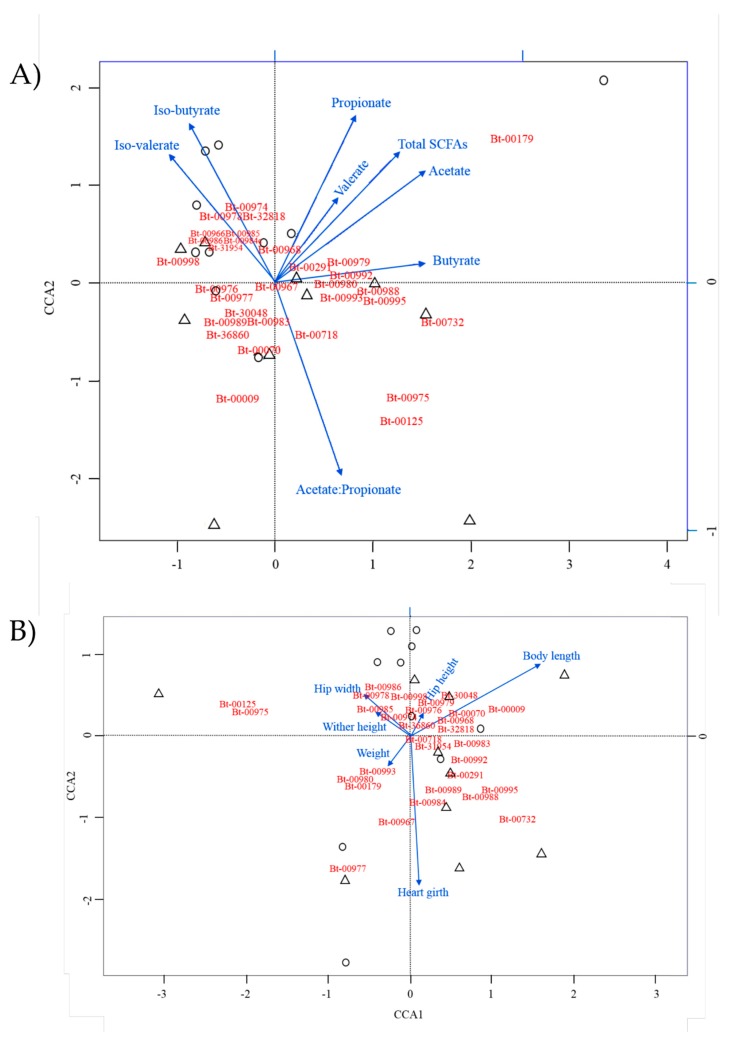
Canonical correspondence analysis (CCA) to uncover associations between main OTUs and short chain fatty acid (SCFA) parameters (**A**) or dairy calf performance attributes (**B**) as explanatory variables. The length of an arrow represents the relative influence of its corresponding explanatory variable on the distribution of the OTUs analyzed. EO and control samples are represented by circles and triangles, respectively.

**Table 1 microorganisms-07-00120-t001:** Relative abundance (%) of main bacterial taxonomic groups in the rumen of dairy calves fed an EO-supplemented or control diet. Values shown represent mean and standard error of the mean, respectively.

Taxonomic Affiliation	EO	Control	*P*-Value
**Firmicutes**	43.68 ± 6.92	73.22 ± 6.79	0.0069 *
Lachnospiraceae	26.87 ± 6.92	51.53 ± 8.44	0.0364 *
Erysipelotrichaceae	12.11 ± 5.65	9.99 ± 4.97	0.7812
Ruminococcaceae	0.91 ± 0.42	3.71 ± 1.74	0.1349
unclassified Clostridiales	1.72 ± 0.59	1.94 ± 0.48	0.7802
Other Firmicutes	2.07 ± 0.28	6.04 ± 1.82	0.0453 *
**Bacteroidetes**	44.63 ± 6.28	13.45 ± 6.02	0.0021 *
Prevotellaceae	44.20 ± 6.27	9.70 ± 5.94	0.0009 *
unclassified Bacteroidales	0.18 ± 0.07	0.06 ± 0.02	0.1359
Other Bacteroidetes	0.26 ± 0.20	5.64 ± 4.08	0.2048
**Proteobacteria**	3.51 ± 1.32	0.25 ± 0.17	0.0246 *
unclassified Gammaproteobacteria	3.49 ± 1.32	0.17 ± 0.13	0.0222 *
Other Proteobacteria	0.02 ± 0.01	0.08 ± 0.05	0.2111
**Actinobacteria**	2.77 ± 1.88	6.37 ± 3.28	0.3531
Coriobacteriales	2.75 ± 1.88	6.27 ± 3.28	0.3638
**Other Bacteria**	5.41 ± 1.70	6.72 ± 3.94	0.7643

* Means showing a statistical difference (*P* < 0.05).

**Table 2 microorganisms-07-00120-t002:** Alpha diversity indices and coverage from ruminal bacterial communities of dairy calves fed an EO-supplemented or control diet. Values are presented as means and standard error of the mean, respectively.

Index	EO	Control	*P*-Value
Chao1	484 ± 48	543 ± 80	0.5375
OTUs *	206 ± 19	219 ± 25	0.6760
Shannon	3.18 ± 0.25	3.27 ± 0.24	0.8052
Simpson	0.16 ± 0.04	0.14 ± 0.03	0.7518
Coverage (%)	91.5 ± 0.8	90.7 ± 1.2	0.5723

* OTUs: operational taxonomic units.

**Table 3 microorganisms-07-00120-t003:** Relative abundance (%) of main operational taxonomic units (OTUs) assigned to Bacteriodetes, Proteobacteria, and Actinobacteria in rumen samples collected from dairy calves fed an EO-supplemented or control diet. Values shown represent mean and standard error of the mean, respectively.

OTUs	EO	Control	*P*-Value	Closest Valid Taxon (id%)
**Bacteriodetes**				
SD_Bt-00966 ^a^	19.51 ± 5.32	2.70 ± 1.80	0.008	*P. ruminicola* (97%)
SD_Bt-00976 ^a^	4.74 ± 1.32	8.01 ± 5.02	0.536	*P. ruminicola* (90%)
SD_Bt-00979 ^a^	2.35 ± 2.10	0.02 ± 0.01	0.281	*P. salivae* (89%)
SD_Bt-00985 ^a^	1.92 ± 1.02	0.18 ± 0.08	0.105	*P. salivae* (89%)
SD_Bt-00986 ^a^	0.91 ± 0.34	0.22 ± 0.13	0.080	*P. ruminicola* (95%)
SD_Bt-32818 ^a^	1.11 ± 0.79	0.08 ± 0.05	0.212	*P. multisaccharivorax* (93%)
Total	**30.55**	**11.22**		
**Proteobacteria**				
SD_Bt-00978 ^b^	3.44 ± 1.30	0.17 ± 0.13	0.022	*Haemophilus influenzae* (84%)
**Actinobacteria**				
SD_Bt-00967	1.96 ± 1.37	4.98 ± 2.69	0.331	*Olsenella umbonata* (99%)

Taxonomic affiliations: ^a^ Prevotellaceae; ^b^ unclassified Gammaproteobacteria.

**Table 4 microorganisms-07-00120-t004:** Relative abundance (%) of main operational taxonomic units (OTUs) assigned to Firmicutes in rumen samples collected from dairy calves fed an EO-supplemented or control diet. Values shown represent mean and standard error of the mean, respectively.

OTUs	EO	Control	*P*-Value	Closest Valid Taxon (id%)
SD_Bt-00009 ^a^	4.15 ± 1.47	7.57 ± 5.85	0.577	*Butyrivibrio hungatei* (91%)
SD_Bt-00070 ^a^	1.74 ± 0.54	1.71 ± 0.97	0.982	*Clostridium aminophilum* (91%)
SD_Bt-00179 ^a^	3.54 ± 3.26	0.26 ± 0.17	0.329	*Lachnospira pectinoschiza* (89%)
SD_Bt-00291 ^a^	0.96 ± 0.55	2.60 ± 1.34	0.271	*Coprococcus catus* (90%)
SD_Bt-00718 ^a^	0.92 ± 0.34	1.41 ± 0.45	0.400	*Eisenbergiella tayi* (92%)
SD_Bt-00968 ^a^	0.84 ± 0.38	2.17 ± 1.46	0.389	*Butyrivibrio fibrisolvens* (90%)
SD_Bt-00977 ^a^	4.51 ± 3.06	0.64 ± 0.43	0.227	*Butyrivibrio fibrisolvens* (91%)
SD_Bt-00980 ^a^	0.52 ± 0.23	6.56 ± 4.38	0.185	*Butyrivibrio fibrisolvens* (89%)
SD_Bt-00983 ^a^	0.05 ± 0.03	3.76 ± 1.95	0.073	*Butyrivibrio fibrisolvens* (91%)
SD_Bt-00988 ^a^	0.48 ± 0.19	1.64 ± 0.85	0.198	*Lachnospira multipara* (91%)
SD_Bt-00993 ^a^	0.27 ± 0.15	1.00 ± 0.63	0.269	*Clostridium bolteae* (87%)
SD_Bt-00998 ^a^	0.15 ± 0.08	1.50 ± 1.39	0.347	*Clostridium lavalense* (90%)
SD_Bt-30048 ^a^	1.08 ± 0.46	1.40 ± 0.71	0.718	*Butyrivibrio fibrisolvens* (91%)
SD_Bt-31954 ^a^	0.50 ± 0.24	1.87 ± 0.94	0.176	*Butyrivibrio fibrisolvens* (90%)
SD_Bt-00974 ^b^	8.85 ± 5.45	1.14 ± 0.55	0.176	*Kandleria vitulina* (89%)
SD_Bt-00975 ^b^	0.48 ± 0.35	5.16 ± 5.00	0.363	*Catenibacterium mitsuokai* (88%)
SD_Bt-00989 ^b^	0.53 ± 0.29	1.71 ± 0.89	0.225	*Eubacterium cylindroides* (92%)
SD_Bt-00992 ^b^	0.62 ± 0.29	1.01 ± 0.50	0.509	*Solobacterium moorei* (91%)
SD_Bt-00125 ^c^	0.03 ± 0.01	1.25 ± 1.21	0.324	*Ruminococcus albus* (90%)
SD_Bt-00995 ^c^	0.63 ± 0.33	1.80 ± 1.29	0.390	*Ruminococcus albus* (86%)
SD_Bt-00732 ^d^	0.08 ± 0.05	1.11 ± 1.05	0.338	*Mogibacterium pumilum* (92%)
SD_Bt-00984 ^d^	1.25 ± 0.59	1.55 ± 0.98	0.797	*Syntrophococcus sucromutans* (91%)
SD_Bt-36860 ^e^	0.12 ± 0.06	1.72 ± 1.00	0.129	*Dialister succinatiphilus* (99%)
Total	**32.94**	**50.63**

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
