# Peer review of "Feeding Essential Oils to Neonatal Holstein Dairy Calves Results in Increased Ruminal Prevotellaceae Abundance and Propionate Concentrations"

_microorganisms, 2019, doi:10.3390/microorganisms7050120_

Round 1
Reviewer 1 Report
The paper is well written and the experiment setup & analysis appears thorough and sound.
I have the following recommendations (ordered by section):
Materials & methods-
Is it necessary to outline 3 feeding strategies (varying concentrations of EO- 0.5X, 1X, 1,5X) if only one treatment was investigated in this study, i.e. 1X EO.
How was the EO administered? When did administration begin (birth? 1 week? etc.) What was the EO% to total dry feed? Was the volume of EO constant throughout the six weeks or did the amount of EO increase coinciding with an increase in age and solid food?
L99-100: What was the average daily feed intake at weaning/sampling? Can this be clarified
L101: What volume of rumen sample was taken from each cow?
L127 - L130: Please elaborate on how sample were prepared for sequencing
L177-L178: If the level of significance has been chose (0.05) recommend you do not use the phrase 'tendency toward significance'. Difference was not deemed significant by the test designed.
Results-
Why was family level picked for analysis? Differences at phylum level? Genus level?
High variability between cows within treatment groups (figure 2), this shouldn't be ignored.
It is surprising that the PCoA and diversity indices do not agree. Considering the significant differences in relative abundances measured between bacteria even at phyla level it is surprising that the beta diversity tests show no significance.
As the bacterial abundance is measured in relative terms, a perceived increase in abundance of an OTU does not necessarily mean the total abundance of this OTU has increased. It is also possible that other bacteria have been suppressed (potentially by the EO) causing the OTU to appear more abundant. Can this be clarified?
L188: Again recommend rephrasing, have deemed p < 0.05 to be the threshold for significance in materials & methods, anything above this is not significant.
L193: Control -> control, change this throughout
Table 1: It would be helpful to bold or place an asterisk beside significant p values to improve visibility.
Section 3.4: Indicate whether the correlations are positive or negative in each case.
Figure 4: This figure is a little difficult to interpret, it may be clearer in another format e.g. a heat map
Discussion-
L305- Too definitive. The extent of our ability to manipulate the microbiome in early life and the persistence of these treatments is still under study and there has been no conclusive evidence thus far that it is possible to 'program' the rumen microbiome in early life. Rephrase to e.g. '...may provide a window...'
L344 - L345: Considering the microbial diversity and functional redundancy present in the rumen, is it likely that a significant change in SCFA profile could be caused by just 2 OTUS?
Were any rumen development or performance measurements taken during this study to further support conclusions that EO supplemented calves will have increased performance?
Author Response
Rebuttal to Reviewer 1
The paper is well written and the experiment setup & analysis appears thorough and sound.
I have the following recommendations (ordered by section):
Thank you for your comprehensive review. We hope that we have been able to address your comments and recommendations to your satisfaction.
Materials & methods-
Rev1- Comment1
Is it necessary to outline 3 feeding strategies (varying concentrations of EO- 0.5X, 1X, 1,5X) if only one treatment was investigated in this study, i.e. 1X EO.
Agreed. We have removed the extra information as recommended by the reviewer. The relevant information has been condensed into one sentence, which reads (lines 79-82 of the revised MS):
“For the purpose of the microbiome study presented in this report, rumen fluid was sampled from a subset of 10 of the animals fed milk replacer supplemented with EO (3.75g/feeding, mixed with the milk replacer), and from 10 of the animals fed milk replacer without supplementation.”
Rev1- Comment2
How was the EO administered? When did administration begin (birth? 1 week? etc.) What was the EO% to total dry feed? Was the volume of EO constant throughout the six weeks or did the amount of EO increase coinciding with an increase in age and solid food?
The answers to these questions are:
· Feeding of EO started 3 days after birth (calves were fed colostrum on the first 2 days)
· A constant dose of EO was mixed with the milk replacer, and the amount was not adjusted based on DMI
· Since the daily amount of EO that was fed remained constant, the percentage of EO to total DMI decreased throughout the trial because DMI increased as the trial progressed.
This information has been integrated in the ‘Sample collection’ section of the Materials and Methods (lines 88-91 of the revised MS):
“Calves were fed colostrum for the first two days after birth. Milk replacer (with or without EO supplementation, according to treatment group) was then offered by bucket feeding twice every day from d3 until d35, then reduced to once every day starting at d36 to facilitate weaning at d42.”
Rev1- Comment3
L99-100: What was the average daily feed intake at weaning/sampling? Can this be clarified
In the week prior to sampling, the average daily DMI was 1.38 kg/d for control-fed calves and 1.45 kg/d for EO-fed calves; these numerical differences were not significant (P > 0.05).
This information is now provided on lines 92-94 of the revised MS:
“From week 1 until week 6, calf starter intake increased from approximately 2% of total dry matter intake to approximately 70%; the average daily DMI in the week prior to sampling was 1.38 kg/d for control calves and 1.45 kg/d for EO-fed calves (P > 0.05) [13]. The amount of EO supplemented per feeding remained constant during the trial, and it was not adjusted as a function of calf growth or increased DMI.”
Rev1- Comment4
L101: What volume of rumen sample was taken from each cow?
The volume collected was 50 ml / sample. This information is now provided on line 97 of the revised MS:
“Separate samples (50 ml / sample) were collected for microbiome and SCFA analysis, with the latter supplemented with 25%. metaphosphoric acid (W/V) at a ratio of 4:1 before freezing.”
Rev1- Comment5
L127 - L130: Please elaborate on how sample were prepared for sequencing
Done. We have added the following passage (Lines 126-129 of the revised MS):
“Briefly, libraries for 16S rRNA amplicons were prepared using overhanging primers targeting the 27F and 519R recognition sequences; the primer overhangs encoded barcodes for sample indexing as well as adapter sequences. MiSeq 2x300 sequencing was performed using the MiSeq Reagent Kit v3 following the manufacturers specifications.”
Rev1- Comment6
L177-L178: If the level of significance has been chose (0.05) recommend you do not use the phrase 'tendency toward significance'. Difference was not deemed significant by the test designed.
Following your recommendation, we have removed “, and a tendency towards statistical significance was indicated when 0.05 < P ≤ 0.10” from the sentence.
The passage now reads (lines 175-176 of the revised MS)
“Means of two groups were considered to be significantly different when P ≤ 0.05.”
Results-
Rev1- Comment7
Why was family level picked for analysis? Differences at phylum level? Genus level?
For the phylum level, Table 1 presented in the original manuscript shows statistically supported differences in composition for Firmicutes, Bacteroidetes and Proteobacteria between the two treatment groups (lines 195-197; this information was provided in the manuscript submitted for review).
For family vs genus, we selected family-level for taxonomic analysis simply because it is more inclusive than genus-level. The rumen remains a poorly characterized microbial ecosystem with an estimated 95% of bacterial species still uncharacterized. Consequently, a high proportion of sequences cannot currently be assigned to known genera. However, we have found that many sequences that belong to unknown genera can be assigned at the family-level, which allows us to minimize to some degree the prominence of taxonomic groups designated as ‘unclassified’.
Rev1- Comment8
High variability between cows within treatment groups (figure 2), this shouldn't be ignored.
Agreed. We have added the following passage to the discussion (lines 342-344 of the revised MS):
“While 16S rRNA-based taxonomic analysis revealed a substantial degree of variation in composition amongst calves in this study, differences between treatments were observed.”
Rev1- Comment9
It is surprising that the PCoA and diversity indices do not agree. Considering the significant differences in relative abundances measured between bacteria even at phyla level it is surprising that the beta diversity tests show no significance.
It is important to note that both analyze different aspects of OTU composition. Alpha diversity indices only take into account the number of OTUs and the OTU abundances, while PCoA also takes into account whether samples share common OTUs.
Rev1- Comment10
As the bacterial abundance is measured in relative terms, a perceived increase in abundance of an OTU does not necessarily mean the total abundance of this OTU has increased. It is also possible that other bacteria have been suppressed (potentially by the EO) causing the OTU to appear more abundant. Can this be clarified?
Yes. We have added the following passage to the discussion (lines 381-384 of the revised MS):
“As the analyses presented in this report were performed using relative abundance, it also remains to be determined whether the observed changes in composition were the result of increased cell numbers for resistant bacterial OTUs or a consequence of growth suppression of susceptible bacterial groups.”
Rev1- Comment11
L188: Again recommend rephrasing, have deemed p < 0.05 to be the threshold for significance in materials & methods, anything above this is not significant.
We have followed your recommendation and rephrased the statement which now reads (lines 186-188 of the revised MS):
“A numerical difference in the acetate : propionate ratio was observed between the two treatments (EO: 1.24 ± 0.04; Control: 1.38 ± 0.06; P = 0.072).”
Rev1- Comment12
L193: Control -> control, change this throughout
Done.
Rev1- Comment13
Table 1: It would be helpful to bold or place an asterisk beside significant p values to improve visibility.
Done.
Rev1- Comment14
Section 3.4: Indicate whether the correlations are positive or negative in each case.
All correlations presented in this section were positive. To clarify, we have added the following text to this section (lines 289-290 of the revised MS):
“Based on Pearson correlation coefficient analysis (Supplementary Table 3), the following positive correlations were identified.”
Rev1- Comment15
Figure 4: This figure is a little difficult to interpret, it may be clearer in another format e.g. a heat map.
Respectfully, we prefer using a triplot to present CCA (as currently provided in figure 4 of the submitted manuscript) rather than with a heat map, because the triplot is better at representing potential associations between OTUs, samples and SCFAs / calf performance parameters.
Discussion-
Rev1- Comment16
L305- Too definitive. The extent of our ability to manipulate the microbiome in early life and the persistence of these treatments is still under study and there has been no conclusive evidence thus far that it is possible to 'program' the rumen microbiome in early life. Rephrase to e.g. '...may provide a window...'
Agreed. The passage now reads (lines 303-305 of the revised MS):
“This stage may provide a window of opportunity for manipulation, potentially allowing to increase the productivity and health of mature host animals through modulating the composition of their developing rumen microbiome [23, 24].”
Rev1- Comment17
L344 - L345: Considering the microbial diversity and functional redundancy present in the rumen, is it likely that a significant change in SCFA profile could be caused by just 2 OTUS?
It was not our intention to suggest that only 2 OTUs would be responsible for these changes, but rather to suggest that these 2 OTUs represent the most likely candidates to be involved.
We have softened the language of this passage, which now reads (line 344-349 of the revised MS):
“Notably, the respective abundances of two OTUs, SD_Bt-00966 and SD_Bt-00978, were found to be significantly greater in the rumen of EO fed calves compared to control calves by a factor of 7.2X and 20.2X. While taking into consideration the level of microbial diversity and functional diversity that exists in the rumen, higher propionate levels in response to EO could involve these OTUs at least in part, perhaps through expression of proprionate production pathways [32, 33].”
Rev1- Comment18
Were any rumen development or performance measurements taken during this study to further support conclusions that EO supplemented calves will have increased performance?
We did not collect these data, so the conclusions remain hypothetical or speculative. Taking this into consideration, we have softened the language of the relevant passage, which now reads (Lines 399-400 of the revised MS):
“In conclusion, the results of this study support a beneficial effect of EO on the SCFA profile of dairy calves, which may potentially promote increased performance later in life.”

Reviewer 2 Report
This is a well-written paper on an interesting topic that should interest readers.
Two small points:
a) In line 340, you might add a short comment on why propionate enhances animal performance (e.g. glucose synthesis)
b) Apart from their effects on microbiota, essential oils such as thymol (which is included in you blend of oils) have also been shown to affect the transport of substrates across a variety of gastrointestinal epithelia, including the rumen. You might wish to discuss this aspect briefly.
Author Response
Rebuttal to Reviewer 2 comments
Comments and Suggestions for Authors
This is a well-written paper on an interesting topic that should interest readers.
Thank you, we very much appreciate your positive feedback.
Two small points:
Rev2- Comment1
a) In line 340, you might add a short comment on why propionate enhances animal performance (e.g. glucose synthesis)
Agreed, thank you for this recommendation. The passage now reads (line 338 of the revised MS):
“While propionate may also be used to a lesser extent as a source of energy for rumen papillae development [31], its main effect would more likely be to improve animal performance, through its role as a major substrate for gluconeogenesis, rather than promoting rumen development.”
Rev2- Comment2
b) Apart from their effects on microbiota, essential oils such as thymol (which is included in you blend of oils) have also been shown to affect the transport of substrates across a variety of gastrointestinal epithelia, including the rumen. You might wish to discuss this aspect briefly.
Thank you for this recommendation. We have added the following sentence to the discussion (lines 384-388 of the revised MS), as well as supporting reference #49:
“In addition to their antimicrobial effects, EO can also affect host gut cells, notably by modulating the activity of membrane channels and the uptake of ions [49]; it thus also remains to be determined whether EO modulation of ruminal bacterial communities and host cell activity represent independent effects or can act in synergy.”
49. Braun, H.-S.; Schrapers, K.T.; Mahlkow-Nerge, K.; Stumpff, F.; Rosendahl, J. Dietary supplementation of essential oils in dairy cows: evidence for stimulatory effects on nutrient absorption. Animal 2019, 13, 518–523.
